# *Oxalis tetraphylla* (Class: Magnoliopsidae) Possess Flavonoid Phytoconstituents with Nematocidal Activity against *Haemonchus contortus*

**DOI:** 10.3390/pathogens11091024

**Published:** 2022-09-08

**Authors:** Ana Yuridia Ocampo-Gutiérrez, Víctor Manuel Hernández-Velázquez, Alejandro Zamilpa, María Eugenia López-Arellano, Agustín Olmedo-Juárez, Rosa Isabel Higuera-Piedrahita, Edgar Jesús Delgado-Núñez, Manasés González-Cortázar, Pedro Mendoza-de Gives

**Affiliations:** 1Laboratory of Helminthology, National Centre for Disciplinary Research in Animal Health and Innocuity (CENID-SAI), National Institute for Research in Forestry, Agriculture and Livestock, INIFAP-SADER, Jiutepec 62550, Mexico; 2Biological Control Laboratory, Biotechnology Research Center, Autonomous University of the State of Morelos, Cuernavaca 62209, Mexico; 3South Biomedical Research Center, Social Security Mexican Institute (CIBIS-IMSS), Xochitepec 62790, Mexico; 4Faculty of Advanced Studies-Cuautitlán, National Autonomous University of Mexico, Cuautitlán 54714, Mexico; 5Faculty of Agricultural, Livestock and Environmental Sciences, Autonomous University of the State of Guerrero, Iguala de la Independencia 40040, Mexico

**Keywords:** nematocidal phytoconstituents, parasitic nematodes, 4-leaf clover, natural compounds

## Abstract

The nematocidal activity of an *Oxalis tetraphylla* hydroalcoholic extract against the nematode *Haemonchus contortus* (*Hc*) was assessed in vitro and the major compounds associated with nematocidal activity were identified. One hydroalcoholic extract was obtained from *O. tetraphylla* stems and leaves (*Ot HE-SLE*). The in vitro lethal concentrations (LC50 and LC90) against both eggs and exsheathed *Hc* infective larvae (L3) were assessed. *Ot HE-SLE* showed a potent ovicidal activity (LC50 = 0.213 mg/mL; LC90 = 0.71 mg/mL) and larvicidal effect (LC50 = 28.01 mg/mL; LC90 = 69.3 mg/mL). Later on, the extract was bipartitioned to obtain an ethyl acetate phase (*EtOAc-Ph*) and an aqueous phase (*Aq-Ph*). Both phases were assessed against *Hc* eggs at 0.25 and 1.0 mg/mL concentrations. The results with *EtOAc-Ph* showed 93.6% ovicidal activity, while 96.6% was recorded with *Aq-Ph* at 48 h post-confrontation (PC). In the case of larvicidal activity, both phases were assessed at 28 mg/mL; *Aq-Ph* showed >80% larvicidal activity 24 and 72 h PC, while *EtOAc-Ph* did not show important activity. HPLC analysis showed the presence of coumaric acid and flavonols. Flavonol compounds were the major compounds and were associated with the nematocidal activity. Additionally, the *Aq-Ph* that showed the highest activity was purified, and the fraction F3 showed the highest nematocidal activity.

## 1. Introduction

One of the main problems affecting the sheep industry worldwide is caused by gastrointestinal parasitic nematodes [1,2]. *Haemonchus contortus* is considered to be the most pathogenic parasite due to its bloodsucking feeding habits in the abomasal mucosa of animals. This feature provokes important blood loss that affects mainly young lambs who can exhibit the following signals: loss of appetite, anorexia, weakness, susceptibility to acquire other diseases and even death [3,4]. The regular administration of chemical anthelmintic drugs is the most common strategy to control sheep haemonchosis and other nematodiases of flocks; however, the frequent and continuous use of such drugs in the animals has triggered an imminent presence of anthelmintic resistance in the parasites, which leads to a worrying inefficacy of most of the drugs commercially available [5,6]. For example, recent studies have reported the following effectiveness in different anthelmintic drugs: avermectin 81.28%, ivermectin 86.49%, albendazole 76.21% and levamisole 96.59% [7]; albendazole 83%, ivermectin 57% [8]; fenbendazole 70% [9].

The use of medicinal plants against parasitic nematodes of ruminants has gained the attention of researchers all over the world since this strategy is based on the plant’s self-defense against parasites in nature [10] and it is a sustainable method of control. The genus Oxalis contains plants from the family Oxalidaceae, having about 900 species. *Oxalis tetraphylla* is a widely spread plant found mainly in Mexico, Southern America and Africa. This species is an annual perennial bulbous plant reaching from 0.1 to 0.3 m high [11] and is characterized by bulbs up to 35–40 mm and four triangular leaflets, frequently showing a purpureal band at the proximal half of the bundles. It has masses of brick pink flowers in spring and summer [12]. This species belongs to a group of plants with medicinal properties, i.e., antifungal, antimicrobial, abortive, analgesic and healing properties [12]. This study was designed to identify the presence of compounds associated with the in vitro nematocidal activity of *O. tetraphylla* against either eggs or infective larvae of *H. contortus*.

## 2. Results

The mean numbers of *H. contortus* hatched eggs and larvae recovered from the in vitro confrontation assay after exposure to *O. tetraphylla* hydroalcoholic extract are shown in Table 1.

In the case of activity of the extracts against larvae, higher concentrations were necessary to reach both 50% and 90% lethal concentrations (Table 2).

The *O. tetraphylla* hydroalcoholic extract from stems and leaves showed important activity against eggs of the parasite: at 0.21 and 0.71 mg/mL for LC50 and LC90, respectively, evaluated 48 h post-confrontation. The LC50 and LC90 for the *EHI* required a very low concentration: 0.21 and 0.71 mg/mL, respectively. Meanwhile, 28 and 69.3 mg/mL of the extract were necessary to achieve 50 and 90% of larval mortality; respectively. Ranges are shown in Table 3.

The egg-hatching inhibition percentages attributed to the effect of both ethyl acetate and aqueous phases: 92.5% and 93% for *Aq-Ph* and *EtOAc-Ph* at 0.25 mg/mL, respectively, are shown in Table 4. Meanwhile, at 1 mg/mL, 96.6% and 93.6% inhibition was recorded for *Aq-Ph* and *EtOAc-Ph*, respectively. Statistical analysis showed that the egg-hatching inhibition attributed to the two phases was statistically different compared to the lethal effect observed in the positive control with ivermectin: 3.46% at 0.25 mg/mL and 87.2% at 1 mg/mL.

The results of the nematocidal effect against *H. contortus* (L3) of the two phases of *O. tetraphylla* hydroalcoholic extracts at 28 mg/mL at the different times assessed are shown in Table 5. It is important to consider that the LC50 calculated for the hydroalcoholic extracts from stems and leaves was considered to establish the concentration in this experiment.

### 2.1. Assessment and Identification SD = Standard Deviation of the Active Nematicidal Fractions from the Aqueous Phase of Oxalis tetraphylla against Haemonchus contortus Eggs and Larvae (L3)

The results of the *H. contortus* egg-hatching inhibition assay after exposure to the fractions obtained from *Aq-Ph* showed values in the range of 96–100% activity for all the fractions at the two concentrations assessed, 0.25 and 1 mg/L. The positive control (ivermectin) caused <86% activity at 1 mg/mL. For the bioactive fraction against *H. contortus* larvae at 10 mg/mL concentration in a 70:30 (dichloromethane/methanol) elution system, the results showed >50% larvicidal activity. Meanwhile, at 30 mg/mL, larvicidal activity ranged from 28.4% to 89.2% in the different elution systems (90:10, 80:20, 70:30, 60:40 and 50:50). The highest larvicidal activity (>80%) was found in the 90:10 and 70:30 elution systems after 48 and 72 h. In the 80:20 and 60:40 elution systems, the peaks of activity were >73% and >58%, respectively (Table 6).

Ivermectin (5 mg/mL) was used as a positive control. Values after ± = standard deviation, n = 3, *p* < 0.05, Tukey: means with a different literal are statistically different. * = ivermectin at 5 mg/mL concentration. The results were previously adjusted subtracting the mortality percentage in control water or DMSO 5%. Bold-highlighted numbers correspond to the highest larval mortality percentages achieved.

### 2.2. Thin-Layer Chromatography Results

TLC plates revealed with flavonoid reagent showing colored bands associated with the presence of flavonoid compounds are shown as Appendix A.

### 2.3. HPLC Results

Analysis of the HPLC chromatograms revealed the presence of flavonoids, of the classes flavones and flavonols, in the hydroalcoholic extract (A); these compounds are identified in the aqueous phase chromatogram (peaks 5, 6 and 7) (B). In the ethyl acetate fraction, coumaric acid was the major compound identified, found at 9.301 min retention time (C); coumaric acid is shown as a standard commercial compound (D) (Figure 1).

### 2.4. Microscopic Analysis

When both eggs and larvae were observed under the microscope after being exposed to the *O. tetraphylla* extracts at different concentrations, some interesting findings were observed (Figure 2), i.e., eggs in the control group showed normal egg morula development (Figure 2A). Likewise, larvae exposed to ivermectin and *Aq-Ph* showed similar changes that included embryonic cell atrophy, separation of the cells from the external eggshell, swelling and a dark and diffuse aspect of the morula cells; the embryonic cells appeared to have a loss of integrity (Figure 2B,C). A recently hatched larva 1 (control group in water) is shown in Figure 2D). Other eggs developed larvae that went partially out of the egg; however, these larvae were not able to completely emerge from the egg, with part of the larva staying outside the egg and another part inside the egg (Figure 2E). In the case of larvae (L1, L2 and L3) after 48 h of exposure to *Aq-Ph*, a thickening in the larval body from side to side with larval displacement, and an empty space between the cuticle and the larval body were observed (Figure 2F). Normal larvae were observed in the control group in water (Figure 2G). Larvae exposed to ivermectin remained motionless and loss of cellular integrity was observed; additionally, some larvae showed some slimming at the anterior extreme with a crushed-like aspect (Figure 2H). Finally, larvae exposed to *Aq-Ph* (30 mg/mL) were swollen and displaced, allowing a space to be seen between the larva and the cuticle (Figure 2I).

## 3. Discussion

### 3.1. Ovicidal and Larvicidal Activity

Searching for natural bioactive compounds from plants to reduce the gastrointestinal parasitic burden in young ruminants is gaining interest from researchers all over the world. The present investigation shows evidence of an important egg-hatching inhibitory effect and even a larvicidal effect against *H. contortus* attributed to bioactive compounds present in the stems and leaves of *O. tetraphylla*. It is interesting that after 48 h of exposure, the crude alcoholic extract at only 0.2 and 0.3 mg/mL exerted an important egg-hatching inhibitory effect close to 60% and 70%, respectively. Likewise, the LC50 results using the same crude extract were highly encouraging, since the LC50 was achieved with only 0.213 mg/mL. It is also interesting that the ovicidal efficacies obtained with the different concentrations of *O. tetraphylla* extract were even higher than that obtained with 0.5 mg/mL ivermectin, one of the most potent commercially available anthelmintics all over the world, and for which, with anthelmintic resistance, has already been reported in many countries [13,14]. The results of the present study show that the phytoconstituents present in *O. tetraphylla* are responsible for high in vitro activity against the eggs and larvae of *H. contortus* that, as we mentioned, is considered one of the most pathogenic parasites affecting small ruminants all over the world. In previous research, an *O. tetraphylla* hydroalcoholic extract from stems and leaves recorded 98.95% larval mortality against *H. contortus* exsheathed larvae (L3) after 48 h of exposure at a 20 mg/mL concentration [15]. Our results corroborate the high mortality obtained with this extract. Beyond these results, we achieved the estimated lethal concentrations (LC50 and LC90) that demonstrate that low concentrations of *O. tetraphylla* fractions as well as the bioactive fraction are sufficient to achieve inhibition of *H. contortus* egg hatching (96–98%). Meanwhile, the larvicidal activity ranged from 81 to 82% using the aqueous phase at 28 mg/mL 24 h post-confrontation.

### 3.2. Analysis of Phytoconstituents

Additionally, our study showed the presence of some phytoconstituents associated with nematocidal activity against *H. contortus* eggs and larvae.

The high efficacy obtained with *O. tetraphylla* subfractions against *H. contortus* eggs offers the opportunity to elucidate the molecules responsible for such an effect and to explore beyond the possibility of establishing its possible effect as a potential anthelmintic in farm animals. On the other hand, it is interesting that we found the presence of flavonoids in the bioactive fractions, since these compounds have been identified in other plant extracts, i.e., *Gliricidia sepium*, and have been associated with ovicidal activity against other nematode parasites of ruminants, for example, *Cooperia punctata* [16]. In another study, the different flavonol compounds obtained from a *Leucaena leucocephala* fraction (eluted in dichloromethane/methanol) showed the following ovicidal activity percentages: quercetin 82.21%, caffeic acid 13.42% and scopoletin 4.37% [17]. In our study, we identified the presence of molecules of coumarin, which presumably is one of the *O. tetraphylla* phytoconstituents responsible for its nematocidal activity. One issue of interest in our results is the fact that the almost 100% lethal effect of our *O. tetraphylla* fractions at only 0.21 mg/mL was obtained against *H. contortus* eggs, and this result is even better than those obtained with other routinely used anthelmintics such as albendazole and ivermectin [18]. However, bioguided studies should be considered in order to corroborate the bioactive compounds of the *O. tetraphylla* extract responsible for the nematocidal activity.

The nematocidal activity of cinnamic acid and six analogues has been recorded against *H. contortus* eggs and larvae by Mancilla-Montelongo et al. (2019) [19]. Regarding the results of the HPLC analysis, important major compounds, i.e., coumaric acid, were found; this compound has been isolated from plants belonging to the Leguminosae family and it has proved to possess larvicidal and ovicidal activity against gastrointestinal parasitic nematodes in bovines and small ruminants, i.e., in an in vitro study, coumaric acid and other flavonols obtained from *Acacia cochliacantha* leaves showed a lethal effect close to 100% against *H. contortus* eggs [20]. These compounds have also been associated with important fungicidal and nematocidal activity [21]. Regarding the other compounds, we observed that our HPLC chromatograms included apigenin and rutin; these and other phenolic compounds, including phenolic acids, flavonoids and coumarins, are reported to be in another widely known medicinal plant commonly used in traditional medicine called chamomile (*Matricaria chamomilla*). This plant has been reported to have a number of therapeutic properties including antioxidant, antibacterial, antifungal, anti-parasitic, insecticidal, anti-diabetic, anticancer and anti-inflammatory effects [22]. Apigenin has also been reported to have nematocidal activity against *Meloidogyne incognita*, one of the most important and devastating plagues of worldwide economically important crops [23]. Likewise, rutin has been reported as a phytoconstituent obtained from several plants with nematocidal activity and has even been used as a positive control in trials to assess in vitro nematocidal activity due to its anthelmintic activity [24].

On the other hand, regarding the nematocidal activity of *O. tetraphylla* fractions against *H. contortus* infective larvae (L3), it is important to remark that, of the five bioactive fractions from the aqueous phase, fraction 3 (eluted with 70:30 dichloromethane/methanol) was the one showing the highest larval mortality, and TLC plates showed the presence of flavonoid compounds (see Appendix A). In the TLC plates, we observed additionally important molecules, i.e., coumaric acid and coumarins that have been reported to have ovicidal and larvicidal activity (mentioned above). Likewise, other plants, i.e., *Combretum mucronatum*, have been shown to possess flavonol molecules, i.e., epicatechins, oligomeric proanthocyanidins and flavonoids that have been associated with anthelmintic activity against the free-living nematode *Caenorhabditis elegans* [25]. Flavonol group compounds have been obtained from other plants belonging to the same taxonomic group and from other groups, and important anthelmintic activity has been recorded (Table 7).

Coumarins have also been reported to have an anthelmintic effect against the Indian soil earthworm *Pheretima posthuma* [28]. More recently, some authors have obtained high in vitro anthelmintic activity against *H. contortus* attributed to the effect of coumarins, i.e., Castillo-Mitre et al. (2017) [20], who reported that a potent active fraction from the plant *Acacia cochliacantha* contains caffeoyl derivates and coumaric compounds with potent ovicidal activity against *H. contortus*. Likewise, von Son-de Fernex et al. (2017) [29] reported high anthelmintic activity of a coumarin molecule identified as 2H-chromen-2-one against another ruminant digestive parasitic nematode, *Cooperia punctata*. These authors stated that this molecule inhibits egg hatching and larval development, with an LC50 of 0.024 ± 0.082 mg/mL.

### 3.3. Microscopic Findings

Regarding the morphological changes found in *H. contortus* eggs after exposure to *O. tetraphylla Aq-Ph*, such as swelling and separation of the eggshell coat from the embryo cells as well as embryonic atrophy, in another study, other morphological changes were observed using scanning electron microscopy (SEM), where the effect of coumarin obtained from a leguminous plant *Gliricidia sepium* was assessed against *Cooperia punctata* eggs. It reported debris attachment, a collapsed eggshell structure and multiple external fractures [29]. Coumarins and coumaric derivates have been reported as potent cell proliferation inhibitors [30]. This effect could be responsible for the embryo cell atrophy in the present study. The morphological changes to larvae identified by microscopy in our study are similar to those reported by Olmedo-Juárez et al. (2020) [31], who assessed the hydroalcoholic extracts obtained from other plants, i.e., leaves of the leguminous plant *Acacia cochliacantha* are reported to have a slimming effect on the anterior part of larvae. The microscopic changes observed in *H. contortus* larvae after exposure to *Aq-Ph* included swelling and displacement of larvae from the larval cuticle; morphological changes in *H. contortus* larvae using isorhamnetin, a secondary metabolite belonging to the flavonoid group obtained from *Prosopis laevigata*, were recently reported by Delgado-Nuñez et al., (2020) [32], including larval deformation with loss of internal organ integrity. Additionally, they reported changes in the larvae’s external cuticle, which appeared rough and straight with wavy formations mainly in their lateral nervous cords along the larval body [32]. Other authors have revealed observations about important cell disorganization, muscle cell degeneration, accumulation of electron full vesicles and alterations to the hypodermis; and, to a minor extent, abnormal condensation of cell chromatin was observed in *H. contortus* and *Trichostrongylus colubriformis* infective larvae exposed to an *Onobrychis viciifolia* acetonic extract using transmission electron microscopy (TEM). The authors mention that this plant contains condensed tannins as well as flavonoid compounds that could be responsible for the damage to nematode larvae [33].

## 4. Materials and Methods

### 4.1. Location

This study was carried out at the Helminthology Laboratory of the National Centre for Disciplinary Research in Animal Health and Innocuity, CENID-SAI-INIFAP, Jiutepec, Morelos, Mexico, and at the Southern Biomedical Research Centre (CIBIS-IMSS), Xochitepec, Morelos, Mexico.

### 4.2. Plant Material

*Oxalis tetraphylla* was collected from the woodland zones at the peak of the mountain called “Cerro de la Luz” or “light’s mountain” at San Juan Tlacotenco village, Tepoztlán municipality, Morelos State, Mexico (Figure 3). Plant material was collected on 4 June 2015, when the plants were flowering. Plant yield (dry weight) was as follows: flowers = 7 g, leaves = 150 g, stems = 200 g and roots = 400 g. The plant was taxonomically identified by the biologist Margarita Aviléz from the Acapatzingo Botanical Center at the National Institute of Anthropology and History of Mexico, Cuernavaca City, Morelos State, Mexico. A specimen of the plant was deposited in the herbarium of this institution (registration number: 2058).

### 4.3. Acquisition of Hydroalcoholic Extracts

Five kilograms of *O. tetraphylla* (stem and leaves) were collected and allowed to dry at room temperature (18–25 °C). Dry material was ground in a mill (Pulvex Plastic, Mexico city, Mexico) to obtain a particle size < 4 mm. Later on, the material was put in a 6 L Erlenmeyer flask, distilled water/ethanol (60:40) was added and the mix was allowed to interact for 24 h at 25 °C. The extract was filtered through Whatman filter paper No. 4 and then was concentrated in a Heidolph Laborota 4000 rotary evaporator (Schwabach, Germany), at 90 rpm at 50–55 °C until total dryness. This process was done in triplicate. Hydroalcoholic extracts of stems and leaves (*Ot HE-SLE*) were lyophilized using Heto Drywinner equipment and finally kept at −20 °C until use. All the solvents used in this extraction were obtained from Sigma-Aldrich (98% purity).

### 4.4. Partition of a Hydroalcoholic Extract

The extract *Ot HE-SLE* was partitioned by liquid/liquid chromatography using immiscible water/ethyl acetate solvents (400 mL each) in water, and then it was bipartitioned with ethyl acetate. Fifty-five grams of *Ot HE-SLE* were used, forming the two phases (aqueous and ethyl acetate). Subsequently, both phases were concentrated using a rotary evaporator at reduced pressure (90 rpm, 55 °C), giving 47 g of the aqueous phase of stem and leaves (*Aq-Ph*) and 20 g of the acetate phase (*EtOAc-Ph*), and were lyophilized to eventually evaluate their nematocidal activity against *H. contortus* eggs and infectious larvae [20].

### 4.5. Chromatographic Fractionation of the Aqueous Phase of Oxalis tetraphylla

Twenty-five grams of *Aq-Ph* were dissolved in 5 mL of methanol and 1 mL of water, and subsequently adsorbed with 30 g of silica gel 60 (Merck). A glass column (18 × 42 cm) was packed with 200 g of normal phase silica gel and stabilized with 100% dichloromethane mobile phase. The absorbed fraction (*Aq-Ph*) was added to the column and chromatographic fractionation was started. An elution system with dichloromethane-methanol starting with an initial gradient of dichloromethane 100% was prepared. The polarity was increased with the use of 10% methanol per each litre. Thirteen fractions were obtained (F1–F13) with the following mixtures of dichloromethane/methanol: F1 (90:10), F2 (80:20), F3 (70:30), F4 (60:40), F5 (50:50), F6 (40:60), F7 (30:70), F8 (20:80), F9 (10:90) and F10–F13 (0:100). Fractions were lyophilized to assess their lethal activity against *H. contortus* eggs and infective larvae (L3).

### 4.6. Thin-Layer Chromatography (TLC)

After aqueous phase fractioning, a thin-layer chromatography analysis was performed. TLC plates were impregned with flavonoid reagent (5 mL polyethylene glycol, 5 mL ethanol, 200 mg of Diphenyl boryloxy ethylamine). Plates were dried for 30 s on a Digital Magnetic Stirrer Hot Plate at 150 °C. Colour bands were visualized at 254 and 365 nm UV.

### 4.7. High-Performance Liquid Chromatography (HPLC) Analysis

Chromatographic analysis of *Ot HE-SLE*, *EtOAc-Ph* and *Aq-Ph* was performed using HPLC equipment. A separation module (Waters 2695) equipped with a photodiode matrix detector (Waters 996) was used and chromatograms were analysed using EmpowerPro software (Waters Corporation, Milford, MA, USA). Compound separation was performed using a Supelcosil LC-F column (4.6 mm × 50 mm; 5 µm particle size; Sigma-Aldrich, Bellefonte, PA, USA). The mobile phase consisted of a 0.5% trifluoroacetic acid aqueous solution (dissolvent A) and acetonitrile (dissolvent B). The gradient conditions were: 0–1 min, 0% B; 2–3 min, 5% B; 4–20 min, 30% B; 21–23 min, 50% B; 24–25 min, 80% B; 26–27 min, 100% B; 28–30 min, 0% B. The flux was maintained at 0.9 mL/min. The injection volume was 10 µL. The photodiode matrix detector was fixed at 280 and 300 nm wavelength to identify phenolic compounds. It is important to mention that the HPLC analysis was performed for only the two phases, organic and aqueous [32].

### 4.8. Acquisition of Haemonchus contortus Eggs and Infective Larvae (L3)

The eggs of the parasite were obtained from a lamb artificially infected with 350 *H. contortus* (L3)/kg. Fifty grams of faeces were collected directly from the rectum of this sheep and eggs were extracted by the NaCl saturation technique; the faeces were mixed in 150 mL of salt (40% commercial NaCl). Egg suspension was transferred to 50 mL Falcon tubes and centrifuged at 600 g for 5 min. Eggs taken from the supernatant were passed through a 0.45 µm sieve and washed (three times) with tap water; 20 mL of the egg suspension were transferred to a 50 mL Falcon tube for its use. Infective larvae (L3) were obtained by coprocultures for which faeces were recovered and coprocultures were elaborated in plastic bowls. After a 1-week incubation period at room temperature (25–28 °C), infective larvae were recovered through the funnel Baermann technique. Larvae were washed using the density gradient technique using 40% sucrose [15]. Larvae exsheathing was achieved using 0.187% sodium hypochlorite. Washed larvae were added to this solution and they remained under these conditions for 7 min. After this time, larvae were washed three times by centrifugation for 2 min at 2500 rpm [34]. The lamb used as an egg-donor animal was maintained under controlled conditions according to the principles of animal welfare and the elimination of unnecessary animal suffering, which are Good Management Practices policies well established at INIFAP. The Norma Oficial Mexicana (Official Mexican Standard) with official rule number NOM-052-ZOO-1995 (http://www.senasica.gob.mx), accessed on 1 January 2016, as well as the Ley Federal de Sanidad Animal (Federal Law for Animal Health) DOF 07-06-2012 (https://www.gob.mx/cms/uploads/attachment/file/118761/LFSA.pdf, accessed on 1 January 2016) were strictly followed and all the procedures performed in this study were carried out in accordance with the ethical standards outlined by INIFAP.

### 4.9. Concentrations Used in Each Hydroalcoholic Extract or Phase of Bipartition from Oxalis tetraphylla Stems and Leaves against Haemonchus contortus

The LC50 and LC90 of *Ot HE-SLE* against *H. contortus* eggs were estimated using the following concentrations: 0.065, 0.1, 0.15., 0.2 and 0.3 mg/mL, while the concentrations used in each of *Aq-Ph* and *EtOAc-Ph* against nematode eggs were 0.25 and 1 mg/mL, respectively. On the other hand, the LC50 and LC90 of *Ot HE-SLE* against *H. contortus* larvae were determined using the following concentrations: 20, 25, 30, 35 and 40 mg/mL. The anthelmintic effect of both *Aq-Ph* and *EtOAc-Ph* from bipartitioning was assessed at 28 mg/mL concentration at 48 h post-confrontation.

### 4.10. Experimental Design

#### 4.10.1. Assessing the Ovicidal Activity of *Oxalis tetraphylla* Extract and Phases against *Haemonchus contortus*

Egg confrontation with extract and phases was performed on 96-well microtitre plates. Fifty microlitres of an aqueous suspension containing 150 *H. contortus* eggs were deposited in one well together with 50 μL of the corresponding extract or phase. Every treatment considered four wells as four replicates (n = 4). The experiment was repeated three times. Two different controls were considered, water (negative control) and ivermectin at 0.25 and 1.0 mg/mL (positive controls). Plates were incubated for 48 h at 25–30 °C. Assays were performed after 48 h of confrontation. Non-hatched eggs or emerged larvae were observed and counted under the microscope (4× and 10× magnifications). The egg--hatching inhibition assay (*EHI*) was performed by determining the egg mortality rate, following the technique described by von Son-de Fernex et al. (2015) [17].

#### 4.10.2. Assessing the Effect of *Oxalis tetraphylla* Extract and Phases against *Haemonchus contortus* Larval Activity

The confrontation of larvae/extract or phases was performed in 96-well microtitre plates. Fifty microlitres of an aqueous suspension containing 200 *H. contortus* infective larvae were deposited in each well (n = 4); likewise, 50 μL of the extract or phase were deposited in the well. Ivermectin at 5 mg/mL was considered as a positive control group, and a group with only water was considered as a negative control group. This experiment was performed in triplicate. Plates were incubated at room temperature (25–30 °C) and assays were carried out at 24, 48 and 72 h post-confrontation. Assays were performed following the description of Olmedo-Juárez et al. (2017) [35].

### 4.11. Statistical Analysis

Data were arcsine-transformed [y = arcsin (sqrt (s/100)] and ANOVA was performed to compare the means of egg-hatching inhibition and larval mortality, followed by the complementary Tukey test (*p* = 0.05). Analyses were performed using the statistical software SAS version 9.0.

### 4.12. Lethal Concentrations (LC50 and LC90)

The lethal concentrations for either eggs or larvae were obtained through the Log10 transformation of the data for egg-hatching inhibition or larval mortality using three consecutive replicates. The results were analysed using the PoloPlus program (version 1.0; LeOra Software Company, Petaluma, CA, USA).

### 4.13. Microscopic Analysis

The microscopic observations of *H. contortus* eggs and larvae were recorded in a set of photographs using a Carl Zeiss Micro-imagen GmbH-37081 light microscope (Gottingen, Germany). To achieve this goal, 10 µL drops were taken from the wells where nematodes were exposed to *Aq-Ph*, then deposited on slides and observed under this light microscope at 10× and 40×. A set of microphotographs was taken to show evidence of possible changes in the morphology of eggs and larvae of the nematodes attributed to the effect of the *Aq-Ph*.

## 5. Conclusions

The results of the present study provide evidence that phases obtained from *O. tetraphylla* stems and leaves contain phytoconstituents that possess anthelmintic activity. The two phases obtained showed egg-hatching inhibition activity with values close to 100%. Flavonol and flavonoid derivative compounds such as coumaric acid possess high nematocidal activity and they could be assessed in future studies searching for natural phytocompounds for controlling haemonchosis in small ruminants.

## Figures and Tables

**Figure 1 pathogens-11-01024-f001:**
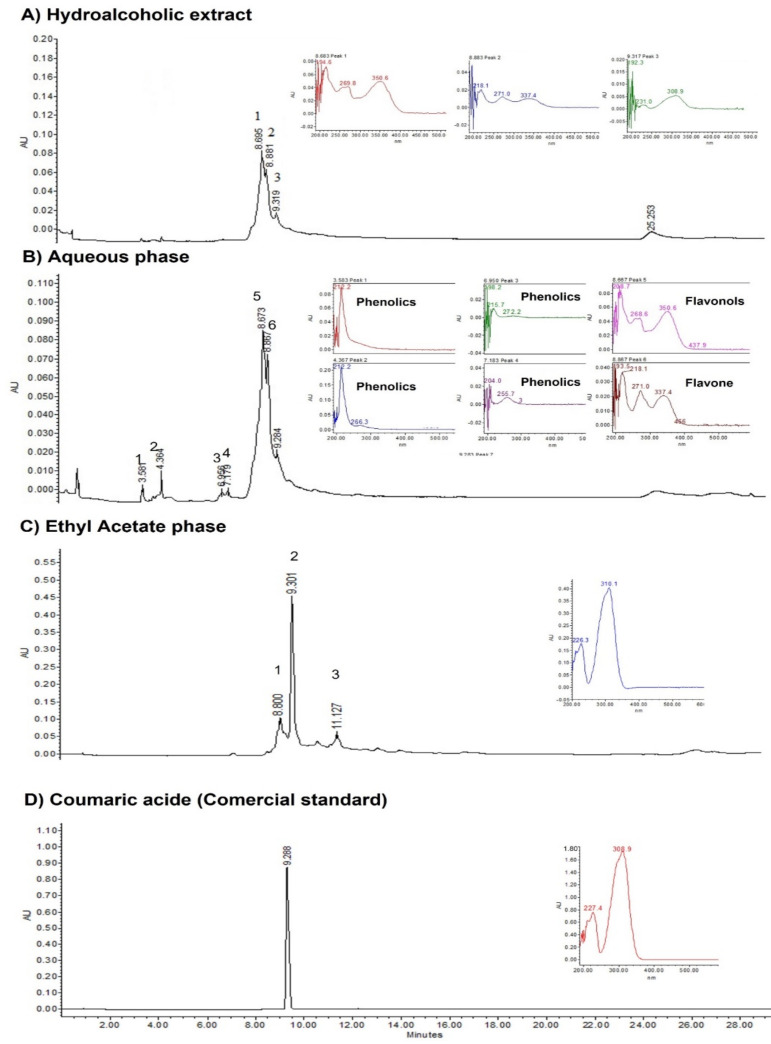
HPLC chromatogram of a hydroalcoholic extract, two phases (an aqueous and an ethyl acetate) from *Oxalis tetraphylla* aerial parts (leaves and stems) and a coumaric acid (commercial standard).

**Figure 2 pathogens-11-01024-f002:**
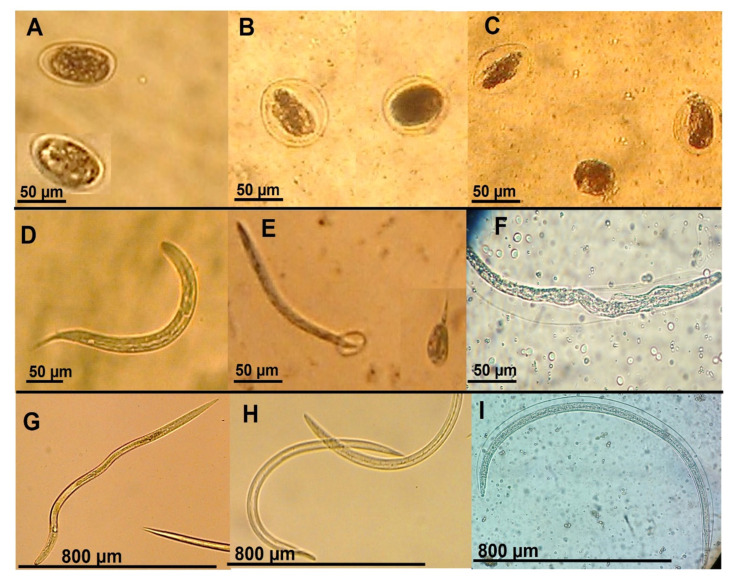
Microscopical analysis of *Haemonchus contortus* eggs and larvae after exposure to an *Oxalis tetraphylla* aqueous phase (*Aq-Ph*) from stem and leaves. (**A**) Normal aspect of eggs (negative control); (**B**) Non-hatched eggs after exposure to ivermectin, showing embryonic atrophy and a dark color diffusion of morula cells; (**C**) Eggs showing swelling and loss of their embryonic cell integrity (embryonic atrophy) after exposure to *Aq-Ph* at 0.25 mg/mL and 1 mg/mL; (**D**) Healthy normal pre-infective larva, recently hatched (negative water control); (**E**) Morphology of two partially hatched larvae; (**F**) A pre-infective larva shown swelling and displacement of the larva from the larva cuticle; (**G**) Healthy infective larvae (negative water control); (**H**) Two motionless larvae after exposure to ivermectin 5 mg/mL; (**I**) Larva showing swelling and displacement of the larva from the larva cuticle after exposure to the *O. tetraphylla Aq-Ph*.

**Figure 3 pathogens-11-01024-f003:**
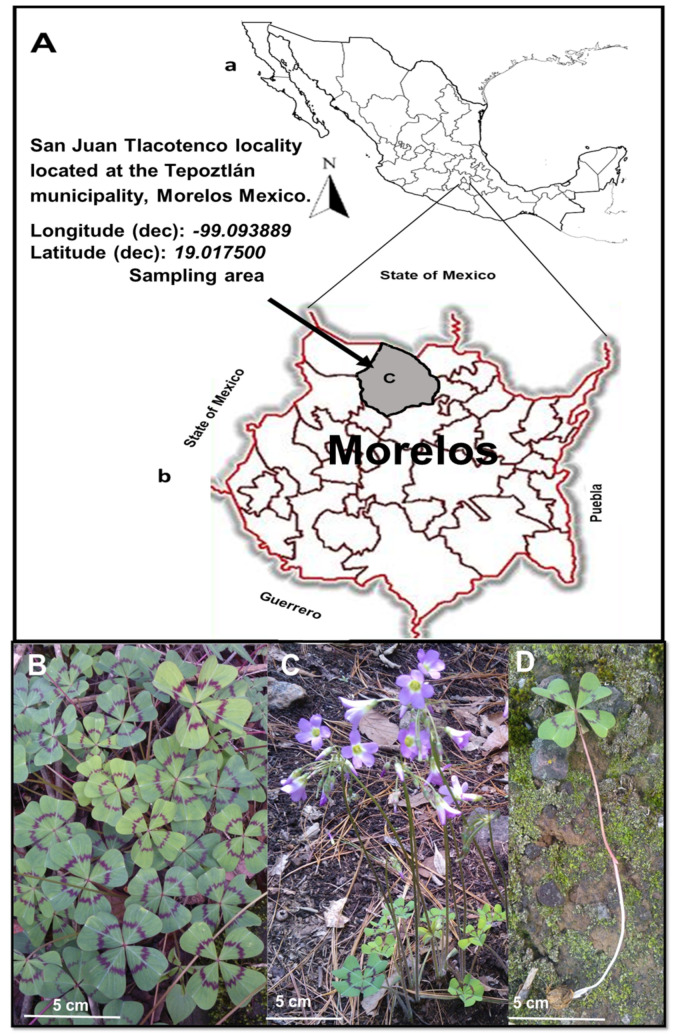
(**A**) Maps showing the site of *Oxalis tetraphylla* plant isolation. (a) Map of the Mexican Republic; (b) Map of the State of Morelos, surrounded by the States of Mexico, Puebla and Guerrero; (c) San Juan Tlacotenco village, the Mountain of Light (El Cerro de la Luz); (**B**) *O. tetraphylla* foliage; (**C**) Plants in flowering stage; (**D**) The whole plant: bulb, stem and leaves.

**Table 1 pathogens-11-01024-t001:** Results of the egg-hatching inhibition (*EHI*) assay showing the means of hatched eggs and L1 larvae recovered after 48 h exposure to an *Oxalis tetraphylla* hydroalcoholic extract against *Haemonchus contortus*.

Concentration (mg/mL)	Mean of Unhatched Eggs/ Mean of Recovered Larvae (*)	*EHI* % ± SD
0.065	22/242	9 ± 2.7 ^e^
0.1	54/197	27.4 ± 1.0 ^de^
0.15	96/238	40.3 ±12.8 ^cd^
0.2	155/266	58.2 ±10.3 ^bc^
0.3	166/241	68.9 ± 8.58 ^b^
Ivermectin (5 mg/mL)	237/237	100 ^a^
Distilled water	18/250	7.2 ± 2.7 ^e^

^a,b,c,d,e^ = Means with a distinct literal are statistically different, * *p* < 0.05; SD = standard deviation.

**Table 2 pathogens-11-01024-t002:** Results of the larval mortality assay showing the means of dead and total recovered larvae after 48 h exposure to five concentrations of *Oxalis tetraphylla* hydroalcoholic extracts from stems and leaves against *Haemonchus contortus* larvae.

Concentration (mg/mL)	Mean of Dead Larvae/Mean of Recovered Larvae (*)	Larval Mortality % ± SD
15	154/417	37 ± 3.7 ^d^
20	196/389	50.3 ± 1.9 ^c^
30	170/308	55.2 ± 2 ^c^
50	335/405	83 ± 2.8 ^b^
60	380/434	88 ± 3.2 ^b^
Ivermectin (5 mg/mL)	380/380	100 ^a^
Distilled water	14/322	4.5 ± 2.75 ^e^

^a,b,c,d,e^ = Means with a distinct literal are statistically different, * *p* < 0.05; SD = standard deviation.

**Table 3 pathogens-11-01024-t003:** Results of *Oxalis tetraphylla* hydroalcoholic extract lethal concentrations 50 and 90 against *Haemonchus contortus* eggs and larvae after 48 h exposure.

Parasitic Stage	Lethal Concentrations and Confidence Interval (Lower–Upper)	
	LC_50_ (mg/mL)	LC_90_ (mg/mL)
Eggs	0.21 (0.18–0.24)	0.71 (0.62–0.97)
Larvae	28 (25.9–29.7)	69.3 (63–78.7)

**Table 4 pathogens-11-01024-t004:** Results about *Haemonchus contortus* egg-hatching inhibition attributed to the effect of two phases of *Oxalis tetraphylla* hydroalcoholic extracts from stems and leaves at two different concentrations.

Treatment	Mean of Unhatched Eggs/Mean of Recovered Larvae (*)	Percentages of Egg-Hatching Inhibition	Mean of Unhatched Eggs/Mean of Recovered Larvae *	Percentages of Egg-Hatching Inhibition ± SD
	0.25 mg/mL		1 mg/mL	
Aqueous phase (*Aq-Ph)*	391/14.3	92.5 ± 7 ^a^	450/2.3	96.6 ± 0.5 ^a^
Ethyl Acetate Phase (*EtOAc-Ph*)	519/0	93 ^a^	603/0	93.6 ^a^
Ivermectin (5 mg/mL)	17/506.6	3.4 ± 3.08 ^b^	442/64.3	87.2 ± 12.8 ^b^
DMSO 0.5%	30/488	6.3 ± 4.41 ^b^	30/488	6.3 ± 4.41 ^c^
Control water	8/392	2.8 ± 3.44 ^b^	8/392	2.8 ± 3.44 ^c^

^a,b,c^ = means with distinct literal into each column are statistically different, *p* < 0.05, * = larvae of the first and second development stages. The results were previously adjusted subtracting the mortality percentage in control water or DMSO, SD = Standard Deviation.

**Table 5 pathogens-11-01024-t005:** Results about the in vitro lethal effect of two phases of *Oxalis tetraphylla* hydroalcoholic extracts from stems and leaves, after three exposure times, expressed as larval mortality percentage.

Treatment	Mean of Dead Larvae/Mean of Live Larvae	% Larval Mortality ±SD	Mean of Dead Larvae/Mean of Live Larvae	% Larval Mortality ±SD	Mean of Dead Larvae/Mean of Live Larvae	% Larval Mortality ±SD
	24 h		48 h		72 h	
Aqueous Phase (*Aq-Ph*)	618/152	82 ± 9.1 ^b^	725/127	82 ± 9 ^b^	747/144	81.6 ± 14 ^b^
Ethyl Acetate Phase (*EtOAc-Ph*)	9/656	1.2 ± 0.5 ^c^	11/690	1.5 ± 0.1 ^c^	8/648	1.7 ± 1.1 ^c^
Ivermectin (5 mg/mL)	211/0	100 ^a^	308/0	100 ^a^	630/0	100 ^a^
DMSO 1%	0/728	0 ^c^	0/760	0 ^c^	760/12	0 ^c^
Distilled water	13/540	3.7 ± 2 ^c^	3/594	0.44 ± 0.4 ^c^	0/620	0 ^c^

^a,b,c^ = means with distinct literal into each column are statistically different, *p* < 0.05. The results were previously adjusted subtracting the mortality percentage in control water or DMSO 1%, SD = Standard Deviation.

**Table 6 pathogens-11-01024-t006:** Percentages of *Haemonchus contortus* infective larvae (L3) mortality after exposure to *Oxalis tetraphylla* stem/leaves aqueous phase fractions at two concentrations and at three lecture times.

Treatment	Mean of Dead Larvae/Mean of Live Larvae	% Larval Mortality ±SD	Mean of Dead Larvae/Mean of Live Larvae	% Larval Mortality ±SD	Mean of Dead Larvae/Mean of Live Larvae	% Larval Mortality ±SD
24 h			48 h		72 h	
**Fractions 10 mg/mL:**						
F1	25/283	8.1 ± 5.6 ^b^	41/264	13.4 ± 7 ^c^	39/245	13.7 ± 2.3 ^c^
F2	5/280	1.7 ± 0.7 ^c^	26/261	9 ± 3.7 ^cd^	10/293	3.3 ± 2.7 ^de^
F3	11/263	4 ± 2 ^bc^	89/202	30.6 ± 3.4 ^b^	142/134	51.4 ± 5 ^b^
F4	5/302	1.6 ± 1.2 ^c^	13/336	3.7 ± 2.4 ^c^	31/292	9.6 ± 3.2 ^cd^
F5	7/302	2.2 ± 0.6 ^bc^	9/336	2.6 ± 2.2 ^c^	10/306	3.1 ± 2.3 ^e^
Ivermectin (5 mg/mL)	338/0	100 ^a^	336/0	100 ^a^	372/0	100 ^a^
Water	0/286	0 ^c^	1/325	0.3 ± 0.6 ^d^	4/297	1.3 ± 1.1 ^e^
**Fractions 30 mg/mL:**						
F1	135/134	50.2 ± 2.5 ^c^	206/46	81.7 ± 12.6 ^b^	242/44	84.6 ± 3.1 ^b^
F2	21/188	10 ± 2.4 ^e^	105/108	49.3 ± 14.4 ^c^	192/68	73.8 ± 5.6 ^bc^
F3	285/45	86.3 ± 4.2 ^b^	328/21	94 ± 2.5 ^b^	280/34	89.2 ± 2.7 ^b^
F4	92/272	25.3 ± 3.3 ^d^	184/174	51.4 ± 5.6 ^c^	194/142	57.7 ± 7.2 ^c^
F5	4/335	1.1 ± 0.9 ^f^	63/274	18.7 ± 6 ^d^	117/295	28.4 ± 9.9 ^d^
Ivermectin (5 mg/mL)	338/0	100 ^a^	336/0	100 ^a^	372/0	100 ^a^
Distilled Water	0/296	0 ^g^	1/325	0.3 ± 0.6 ^e^	4/352	1.1 ± 0.9 ^e^

^a,b,c,d,e,f,g^ = means with distinct literal into each column are statistically different, *p* < 0.05, SD = Standard Deviation.

**Table 7 pathogens-11-01024-t007:** Flavonol group compounds obtained from other plants belonging to the same taxonomic group with important anthelmintic activity.

Plant	Compound	Parasite	Concentration	Effect	Authors
*Onobrychis viciifolia*	Condensed tannin <2000 Da, 3 flavonol glycosides as: rutin, nicotiflorin and narcissin	*Haemonchus contortus* (L3)	1200 µg/mL each compound	Inhibition of larval migration	Barrau et al., 2005 [26]
*Artemisia campestris*	Ethanolic extract: Quercetin, 3 methyl-quercetin, 7 methyl-taxifolin 7,3 methyl-kaempferol. Aqueous extract: Apigenin-6,8-di C-glucoside Eupafolin-glucoside Acethyl-luteolin-glucuronide Apigenin-7-glucoronide Hesperidin Luteolin	*H. contortus* (L3)	Ethanolic extract: 0.5 1 and 2 mg/mL Aqueous extract. 0.5 1 and 2 mg/mL	70–74% Mortality at 24 h. 100% mortality at 24 h	Akkari et al., 2014 [27]
*Acacia cochliacantha*	Caffeoyl and coumaroyl derivatives: caffeic acid, *p*-coumaric acid, ferulic acid, methyl caffeate, methyl-*p*-coumarate, methyl ferulate and quercetin	*H. contortus* eggs	1 mg/mL	Ovicidal activity 71–98%	Castillo-Mitre et al., 2017 [20]

## Data Availability

The data presented in this study are available upon request from the corresponding author.

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
