# Peer review of "Oxalis tetraphylla (Class: Magnoliopsidae) Possess Flavonoid Phytoconstituents with Nematocidal Activity against Haemonchus contortus"

_pathogens, 2022, doi:10.3390/pathogens11091024_

Round 1

Reviewer 1 Report

Dear Authors,

The title of the article sounds interesting. The topic of the manuscript is relevant. The amount of data is sufficient.

The experiments are reproducible. Results presented in a clear authoritative reliable form.

Presented conclusions are the result of the analysis carried out

The experimentum design and data analysis are appropriate, the introduction and discussion are consistent.

But:

- In my opinion introduction could be expanded to include the level of effectiveness with drugs

-Please read the MS once more and correct any minor shortcomings, e.g. punctuation, etc., italics for Latin names

-please look at the literature references, e.g. record 30

Author Response

Reviewer 1,

The title of the article sounds interesting. The topic of the manuscript is relevant. The amount of data is sufficient. 

The experiments are reproducible. Results presented in a clear authoritative reliable form.

Presented conclusions are the result of the analysis carried out

The experimental design and data analysis are appropriate, the introduction and discussion are consistent.

But: 

- In my opinion introduction could be expanded to include the level of effectiveness with drugs

Authors response/actions:

Authors agree with the comment of Reviewer 1. In the new version in the Introduction section of the manuscript we have inserted a paragraph with information about the level of effectiveness of some anthelmintic drugs of common use and their corresponding cites.

-Please read the MS once more and correct any minor shortcomings, e.g. punctuation, etc., italics for Latin names.

-please look at the literature references, e.g. record 30

Authors response/actions:

Authors have carefully checked the MS and we found many mistakes that we finally fixed.

We also corrected the error in  reference number 30.

Reviewer 2 Report

Although this manuscript reports meaningful data concerning Oxalis tetraphylla hydroalcoholic extract against the nematode Haemonchus contortus in vitro and detecting the major active compounds. However, it cannot be accepted for publication by reason of the following aspects requiring circumstantial revision.

1.     Both title and abstract of the paper believe that the main components of Oxalis tetraphylla extract are flavonoid compounds. But the manuscript only provides UV characteristic absorption in HPLC for identification, which may be a wrong conclusion. The characteristic UV absorption of flavonoids is around 260 and 340 nm, but it does not mean that the compounds with this UV absorption are flavonoids. Therefore, this conclusion cannot be directly drawn from the UV absorption spectrum.

2.     The authors presume that coumaric acid and flavonols in the extract may be the main active components in manuscript. But in fact the nematicides of coumaric acid and common flavonols have been reported, and their LC50 is much lower than that of avermectin. Therefore, the strong nematicidal activity of the extract cannot be explained. So, I think more detection and analysis data such as LC-MS should be provided to verify the author's conclusion.

Author Response

Reviewer 2

 Suggestions for Authors

 Although this manuscript reports meaningful data concerning Oxalis 
 tetraphylla hydroalcoholic extract against the nematode Haemonchus 
 contortus in vitro and detecting the major active compounds.
 However, it cannot be accepted for publication by reason of the
 following aspects requiring circumstantial revision.

 Comment 1.Both title and abstract of the paper believe that the main 
 components of /Oxalis tetraphylla /extract are flavonoid compounds. 
 But the manuscript only provides UV characteristic absorption in HPLC
 for identification, which may be a wrong conclusion. The 
 characteristic UV absorption of flavonoids is around 260 and 340 nm, 
 but it does not mean that the compounds with this UV absorption are 
 flavonoids. Therefore, this conclusion cannot be directly drawn from 
 the UV absorption spectrum.

 Authors:

 Dear Reviewer, thank you for your kind comment.

On this regard, the identification of phenolic compounds ie., 
 flavonoids in our lab, is not only based only in the UV spectra. We have 
 established the following procedure: we firstly, perform the thin 
 layer chromatography assay and we visualize and analyze the 
 colorimetric band profile and we compare with standards of reference 
 as a routine procedure. Then, the presence of flavonoids is 
 corroborated by high performance liquid resolution  (HPLC) based on 
 retention times and UV spectra according to standards of reference 
 (Wagner, H. and Bladt, S. (1996) Plant Drug Analysis: A Thin Layer Chromatography Atlas. 2nd Edition, Springer-Verlag, Berlin). http://dx.doi.org/10.1007/978-3-642-00574-9). In the new version of the manuscript in Materials and Methods section (Issue 4.6) we have additionally included a brief description the of the Thin Layer Chromatography procedure we performed. We have also mentioned in Results section a set of TLC photographs showing the presence of flavonoids and we have included these photographs as Supplementary Material (Supplementary Figure 1).

 Comment 2.The authors presume that coumaric acid and flavonols in the
 extract may be the main active components in manuscript. But in fact
 the nematicides of coumaric acid and common flavonols have been
 reported, and their LC-50 is much lower than that of avermectin.
 Therefore, the strong nematicidal activity of the extract cannot be
 explained. So, I think more detection and analysis data such as LC-MS
 should be provided to verify the author's conclusion.

 Authors:

 Regarding your comment about a high nematocidal activity of pure 
 molecules of either coumaric acid or flavonols previously reported 
 has been obtained with a much lower LC-50, even lower than
ivermectin. We think you´re right. However, in our study we used a
 full extract without fractioning, this could explain why our LC-50
 was higher than the ones reported with pure molecules.

We do believe that if we obtained a high nematocidal activity using a full extract; if we had used pure molecules from this extract we could have obtained a much higher nematocidal activity with a lower LC-50.

We hope reviewers feel satisfied with our answers. However, we will be alert to any change or suggestion from reviewers to our manuscript.

Round 2

Reviewer 2 Report

The revised version was improved and I would recommend this manuscript to accept for publication.